# Synergistic Antioxidant Effects of C3G-Enriched *Oryza sativa* L. cv. RD83 Extract and α-Tocopherol Against H_2_O_2_-Induced Oxidative Stress in SH-SY5Y Cells

**DOI:** 10.3390/ijms26136490

**Published:** 2025-07-05

**Authors:** Nootchanat Mairuae, Nut Palachai

**Affiliations:** Biomedical Research Unit, Faculty of Medicine, Mahasarakham University, Mahasarakham 44000, Thailand; nootchanat.m@msu.ac.th

**Keywords:** cyanidin-3-glucoside, *Oryza sativa* L., α-tocopherol, oxidative stress, SH-SY5Y cells, Nrf2 signaling, HDAC1, antioxidant enzymes, neuroprotection, hydrogen peroxide, synergistic effect, epigenetics

## Abstract

Oxidative stress, which contributes to neuronal cell dysfunction, is a critical factor in the pathogenesis of neurodegenerative diseases. Anthocyanins and α-tocopherol have shown potential in mitigating oxidative damage, and their combination may provide synergistic effects. This study investigated the combined effects of a cyanidin-3-glucoside (C3G)-enriched extract derived from *Oryza sativa* L. cv. RD83 and α-tocopherol (C3GE) on hydrogen peroxide (H_2_O_2_)-induced oxidative stress in SH-SY5Y cells. Cells were treated with C3GE during exposure to 200 µM H_2_O_2_. Cell viability, intracellular reactive oxygen species (ROS), and oxidative stress biomarkers, including the activities of superoxide dismutase (SOD), catalase (CAT), and glutathione peroxidase (GSH-Px), as well as malondialdehyde (MDA) levels, were evaluated. Protein expression levels of histone deacetylase 1 (HDAC1), nuclear factor erythroid 2 related factor 2 (Nrf2), heme oxygenase 1 (HO-1), and SOD1 were also assessed. The combined treatment markedly improved cell viability, suppressed ROS accumulation, enhanced antioxidant enzyme activities, and significantly reduced MDA levels, suggesting effective protection against oxidative damage. Mechanistically, C3GE downregulated HDAC1 expression while upregulating Nrf2, HO-1, and SOD1, indicating that its antioxidant and neuroprotective effects are mediated, at least in part, through epigenetic modulation of redox-related signaling pathways. These results demonstrate a synergistic interaction between C3G and α-tocopherol that enhances cellular antioxidant defenses and supports redox homeostasis. In conclusion, the C3GE combination offers a promising therapeutic approach for preventing or attenuating oxidative stress-induced neuronal injury, with potential relevance for the treatment of neurodegenerative disorders.

## 1. Introduction

Oxidative stress, resulting from an imbalance between the production of ROS and the antioxidant defense system, plays a pivotal role in the pathogenesis of neurodegenerative diseases such as Alzheimer’s disease and Parkinson’s disease [1]. Excessive ROS generation leads to lipid peroxidation, protein oxidation, mitochondrial dysfunction, and DNA damage, ultimately triggering neuronal cell death and progressive neurodegeneration [2,3].

Emerging evidence highlights the significance of epigenetic regulation in controlling cellular responses to oxidative stress. Among these, histone modifications, particularly those mediated by histone deacetylases (HDACs), have been shown to repress antioxidant gene expression [4,5]. By contrast, inhibition of HDAC1 has been associated with enhanced activation of the Nrf2 pathway, a key transcription factor that regulates the expression of antioxidant enzymes such as HO-1 and SOD1 [6]. Therefore, targeting epigenetic mechanisms offers a promising strategy for enhancing endogenous antioxidant responses.

Natural antioxidants are increasingly recognized for their neuroprotective potential due to their favorable safety profiles and their ability to mitigate oxidative stress, a major contributor to neurodegeneration. C3G, a predominant anthocyanin in *Oryza sativa* L., has demonstrated potent antioxidant, anti-inflammatory, and neuroprotective properties [7]. Beyond its direct ROS scavenging capacity, C3G preserves mitochondrial function and activates endogenous antioxidant pathways such as Nrf2/ARE signaling [8]. Notably, previous studies have shown that C3G reduces β-amyloid-induced toxicity, improves cognitive function, and attenuates oxidative and inflammatory markers in Alzheimer’s disease models, as well as protects dopaminergic neurons in Parkinson’s disease models [9,10]. Likewise, α-tocopherol, the most bioactive form of vitamin E, acts as a lipophilic antioxidant that safeguards neuronal membranes from lipid peroxidation and modulates redox-sensitive signaling pathways [11]. Its neuroprotective effects have been documented in both experimental and clinical studies involving Alzheimer’s and Parkinson’s diseases, where it reduced oxidative injury and slowed disease progression [12,13]. These findings support the relevance of both compounds as potential therapeutic agents in neurodegenerative contexts.

While several *Oryza sativa* cultivars have been reported to support redox homeostasis and suppress inflammatory responses, the antioxidant and neuroprotective effects of the non-glutinous rice variety *Oryza sativa* L. cv. RD83 (*Mali Dam Nong Khai 62*) remain largely unexplored. This gap highlights the need to assess its therapeutic potential, particularly in combination with other complementary bioactive agents such as α-tocopherol.

Although C3G and α-tocopherol have each demonstrated substantial antioxidant activity, their co-administration may produce enhanced protective effects through synergistic interactions. Such combinations can amplify therapeutic efficacy and potentially overcome challenges related to limited bioavailability or insufficient potency observed with individual compounds [14,15,16,17]. Despite growing interest in synergistic antioxidant strategies, the combined effects of C3G-enriched extract from *Oryza sativa* L. cv. RD83 and α-tocopherol, especially regarding their roles in modulating antioxidant signaling via epigenetic mechanisms in neuronal cells, remain insufficiently studied.

Therefore, this study aims to investigate the synergistic antioxidant effects of C3GE against H_2_O_2_-induced oxidative stress in human neuroblastoma SH-SY5Y cells. Special emphasis is placed on the role of the HDAC1/Nrf2/HO-1 and SOD1 signaling axis to elucidate the underlying epigenetic mechanisms contributing to neuroprotection. This research not only provides novel insights into the potential of combining plant-derived bioactive compounds with essential micronutrients but also contributes to the growing body of evidence supporting the development of food-based strategies for promoting brain health and preventing oxidative stress-related disorders.

## 2. Results

### 2.1. Anthocyanin Content of Oryza sativa L. cv. RD83 Extract

Figure 1A shows the grains of the non-glutinous purple rice variety *Oryza sativa* L. cv. RD83 (*Mali Dam Nong Khai 62*), while Figure 1B illustrates the molecular structure of C3G, the predominant anthocyanin present in its extract. Figure 2 and Table 1 present the high-performance liquid chromatography (HPLC) fingerprint of C3G in the extract of *Oryza sativa* L. cv. RD83.

The analysis revealed that 100 g of dry extract contained a total anthocyanin content of 527.00 ± 8.82 mg. Notably, C3G was identified as the main anthocyanin component in the extract, with a concentration of 364.44 ± 0.01 mg per 100 g of dry extract, accounting for 69.15% of the total anthocyanin content.

### 2.2. Synergistic Effect of the Combination of C3G-Enriched Extract from Oryza sativa L. cv. RD83 and α-Tocopherol

The antioxidant properties of the test samples were evaluated and are summarized in Table 1. The combination formulation (C3GE) demonstrated markedly higher antioxidant activities across all assays, DPPH, FRAP, and ABTS, when compared to either the *Oryza sativa* L. cv. RD83 extract or α-tocopherol alone. The improvements were statistically significant (*p* < 0.001 vs. *Oryza sativa* L. cv. RD83 extract; *p* < 0.01 vs. α-tocopherol), indicating a substantial enhancement of antioxidant capacity through combination.

To explore the nature of the interaction between the two components, combination index (CI) values were calculated for each assay and are shown in Table 2. The CI values for DPPH (0.52 ± 0.04), FRAP (0.69 ± 0.06), and ABTS (0.66 ± 0.02) were all below 1, signifying a synergistic interaction.

In addition, dose reduction index (DRI) values were determined to assess the extent of dose efficiency when the compounds were combined. The combination led to notable reductions in the required doses of each component to achieve antioxidant effects. Specifically, the DRIs for the *Oryza sativa* L. cv. RD83 extract were 7.04 ± 0.53 (DPPH), 4.11 ± 0.45 (FRAP), and 8.44 ± 0.02 (ABTS), while for α-tocopherol, the corresponding DRIs were 2.79 ± 0.22, 2.42 ± 0.23, and 1.87 ± 0.06, respectively.

These results indicated that co-administration of the C3G-enriched extract from *Oryza sativa* L. cv. RD83 and α-tocopherol not only improved antioxidant efficacy but also allowed for lower effective doses of both components. The synergistic behavior observed across multiple assays supported the potential of this combination as a promising strategy for antioxidant-based interventions.

### 2.3. Cytotoxicity and Cytoprotective Efficacy of C3GE

To investigate the neuroprotective capability of C3GE, a cellular model of oxidative stress was established using differentiated SH-SY5Y neuroblastoma cells. Exposure to H_2_O_2_ was employed to simulate oxidative injury, a condition known to disrupt cellular redox balance and initiate apoptotic cascades.

A concentration–response assessment was initially conducted to define the threshold of oxidative challenge. Cells were incubated with H_2_O_2_ concentrations ranging from 0 to 800 µM for 24 h. Among these, 200 µM was selected as the critical point that induced measurable oxidative perturbation while maintaining a viable population for downstream analysis [14].

The next phase focused on evaluating the biocompatibility of C3GE. SH-SY5Y cells were treated with escalating concentrations of the compound (0–1280 µg/mL) for 24 h. As displayed in Figure 3, a gradual reduction in cell viability was noted with increasing concentration. Minimal cytotoxicity was observed at lower concentrations (≤40 µg/mL), while more pronounced declines in viability became evident at 80 µg/mL (*p* < 0.05) and significantly at concentrations ≥ 160 µg/mL (*p* < 0.001). The viability at the highest dose tested fell to 54.90 ± 0.84%.

From these results, 20 µg/mL and 40 µg/mL were identified as optimal working concentrations for subsequent experiments. These doses provided a favorable safety profile and were thus selected to evaluate whether C3GE could attenuate H_2_O_2_-induced cellular injury in this neuronal cell model.

The cytoprotective potential of C3GE was evaluated using SH-SY5Y cells subjected to oxidative stress. As shown in Figure 4, exposure to 200 µM H_2_O_2_ for 24 h significantly reduced cell viability compared to the untreated control group (*p* < 0.001), confirming oxidative injury. Co-treatment with C3GE at both 20 µg/mL and 40 µg/mL markedly attenuated H_2_O_2_-induced cytotoxicity. Both concentrations significantly improved cell viability relative to the H_2_O_2_-only group (*p* < 0.001), suggesting that the combination of C3G and α-tocopherol may enhance cellular defense mechanisms under oxidative conditions.

### 2.4. Efficacy of C3GE on Oxidative Stress Markers

To assess the antioxidant efficacy of C3GE, key markers of oxidative stress, including MDA, a byproduct of lipid peroxidation, and ROS, indicators of oxidative burden, were quantified. As shown in Figure 5 and Figure 6, cells exposed to H_2_O_2_ and vehicle exhibited significantly elevated levels of both MDA and ROS compared to the untreated control group (*p* < 0.001), confirming the induction of oxidative stress.

C3GE treatment significantly attenuated these effects. MDA levels were notably reduced at both 20 µg/mL (*p* < 0.001) and 40 µg/mL (*p* < 0.01), while ROS generation was significantly suppressed at both doses (*p* < 0.001). These results supported the role of C3GE in mitigating oxidative damage by effectively reducing lipid peroxidation and intracellular ROS accumulation.

### 2.5. Effect of C3GE on HDAC1-Mediated Epigenetic Modulation

HDAC1 is a key regulator of gene expression involved in oxidative stress responses [4,5]. To evaluate the epigenetic impact of C3GE, HDAC1 expression was assessed in SH-SY5Y cells following H_2_O_2_ exposure. As shown in Figure 7, H_2_O_2_-treated cells displayed a significant upregulation of HDAC1 expression compared to the untreated control (*p* < 0.001), indicating a stress-induced epigenetic alteration. Notably, treatment with C3GE effectively suppressed HDAC1 expression in a dose-dependent manner. A significant reduction was observed at 20 µg/mL (*p* < 0.01), with a more pronounced effect at 40 µg/mL (*p* < 0.001), suggesting that C3GE may counteract oxidative stress through HDAC1-mediated epigenetic regulation.

### 2.6. Efficacy of C3GE on the Nrf2/HO-1 and SOD1 Pathways

Nrf2 is a critical transcription factor that regulates cellular antioxidant defense mechanisms. Among its downstream targets are HO-1, which plays a role in cytoprotection and anti-inflammatory responses, and SOD1, a key enzyme responsible for dismutating superoxide radicals [6]. Given the observed downregulation of HDAC1 by C3GE, its influence on the Nrf2/HO-1 and SOD1 signaling pathways was further examined.

As depicted in Figure 8, Figure 9 and Figure 10, exposure to H_2_O_2_ led to a significant reduction in the expression levels of Nrf2, HO-1, and SOD1 compared to the untreated control group (*p* < 0.001), confirming oxidative impairment of the antioxidant defense system. Treatment with C3GE significantly reversed these effects. Specifically, Nrf2 expression was upregulated at both 20 µg/mL (*p* < 0.001) and 40 µg/mL (*p* < 0.05). Moreover, HO-1 and SOD1 levels were markedly elevated at both concentrations of C3GE (*p* < 0.001 for all), indicating that the combination treatment restored the antioxidant signaling pathways disrupted by oxidative stress.

### 2.7. Efficacy of C3GE on the Antioxidant Defense Enzyme Activities

To further validate the activation of antioxidant signaling pathways, the enzymatic activities of CAT, SOD, and GSH-Px were measured. As shown in Figure 11, cells exposed to H_2_O_2_ and vehicle displayed significant reductions in CAT and SOD activities (*p* < 0.001) as well as GSH-Px activity (*p* < 0.01) compared to the untreated control group, indicating compromised antioxidant defense.

By contrast, treatment with C3GE markedly improved enzymatic activities. Both concentrations of C3GE significantly increased CAT and GSH-Px activities (*p* < 0.01 for all doses) relative to the H_2_O_2_-treated group. Notably, SOD activity was significantly restored in all C3GE-treated groups (*p* < 0.001), demonstrating a robust enhancement of enzymatic antioxidant defense.

These findings suggested that C3GE effectively restores antioxidant enzyme function disrupted by oxidative stress, supporting its role in cellular protection.

## 3. Discussion

This study provides strong evidence that the combined treatment with C3GE confers neuroprotective effects against H_2_O_2_-induced oxidative stress in SH-SY5Y cells. The protective outcomes were mechanistically linked to the modulation of the HDAC1/Nrf2/HO-1 and SOD1 signaling axis, which plays a pivotal role in controlling redox homeostasis and limiting neuronal injury.

Numerous studies have implicated dysregulated oxidative stress pathways in the pathogenesis of neurodegenerative diseases. In Alzheimer’s disease, overexpression of HDAC1 represses the genes involved in DNA repair and neuronal survival, while impaired Nrf2 activity exacerbates oxidative neuronal damage and neuroinflammation. In Parkinson’s disease, reduced Nrf2 signaling accelerates dopaminergic neuron loss, and heightened HDAC1 activity contributes to mitochondrial dysfunction and α-synuclein aggregation [4,5,6].

Oxidative stress in neuronal cells is primarily driven by an imbalance between excessive ROS production and insufficient antioxidant defenses. The accumulation of ROS, such as superoxide anions (O_2_•^−^) and H_2_O_2_, leads to lipid peroxidation, DNA damage, and protein oxidation, events that contribute to the pathogenesis of various neurodegenerative diseases [3,18]. In our model, H_2_O_2_ exposure elevated intracellular ROS and MDA levels while concurrently depleting antioxidant enzyme activities, including SOD, CAT, and GSH-Px. These changes are characteristic of oxidative damage-induced cellular dysfunction. Notably, C3GE treatment restored redox balance by suppressing ROS accumulation, reducing MDA levels, and enhancing the activity of endogenous antioxidants, suggesting robust attenuation of oxidative stress.

At the mechanistic level, the observed antioxidant effects were strongly associated with activation of the Nrf2 signaling pathway. Under basal conditions, Nrf2 is sequestered in the cytoplasm by Kelch-like ECH-associated protein 1 (Keap1) and targeted for proteasomal degradation [19]. Upon oxidative stress or pharmacological stimulation, Nrf2 escapes Keap1-mediated repression, translocates to the nucleus, and binds to antioxidant response elements (AREs) to initiate the transcription of cytoprotective genes such as HO-1 and SOD1 [20,21,22]. Our study showed that C3GE promoted the nuclear translocation of Nrf2 and significantly upregulated its downstream targets, HO-1 and SOD1, thereby enhancing cellular antioxidant capacity.

Importantly, this activation was closely associated with a significant reduction in HDAC1 expression. HDAC1 is a class I histone deacetylase that plays a central role in chromatin remodeling and gene silencing. Emerging evidence suggests that HDAC1 acts as a negative regulator of Nrf2 by promoting chromatin condensation at ARE-containing promoters, thereby suppressing antioxidant gene expression [23,24,25]. By inhibiting HDAC1, C3GE may facilitate a more open chromatin configuration, allowing for greater transcriptional access and enhanced Nrf2-driven gene expression. This epigenetic mechanism likely contributes to the robust induction of HO-1 and SOD1 observed in the co-treatment condition, highlighting the potential of C3GE to orchestrate multiple levels of antioxidant defense.

HO-1 is a stress-inducible enzyme that catalyzes the degradation of heme into biliverdin, free iron, and carbon monoxide, all of which possess antioxidant and anti-inflammatory properties [26,27,28]. By upregulating HO-1, C3GE may help to mitigate oxidative injury not only through direct ROS scavenging but also by modulating the redox-sensitive signaling environment. Similarly, SOD1 catalyzes the dismutation of superoxide radicals into H_2_O_2_, which is subsequently detoxified by CAT and GSH-Px [29,30]. The concurrent upregulation of SOD1 protein expression and the restored activities of CAT, SOD, and GSH-Px in our study indicates a coordinated activation of the enzymatic defense network, likely orchestrated by Nrf2.

Moreover, the reduction in intracellular ROS levels following C3GE treatment underscores the functional relevance of this pathway. Lower ROS levels reduce the burden on cellular membranes, as reflected by the decreased MDA levels, a marker of lipid peroxidation. The restoration of antioxidant enzyme activities and reduction of MDA levels suggest that C3GE not only prevents ROS accumulation but also supports membrane integrity and mitochondrial function, critical for neuronal survival under oxidative stress conditions.

The synergistic interaction between C3G and α-tocopherol can be rationalized by their complementary biochemical properties, distinct cellular localizations, and mutually reinforcing mechanisms of antioxidant defense. C3G, a water-soluble anthocyanin, is known for its potent free radical scavenging ability, particularly in aqueous environments. It exerts antioxidant effects not only by directly neutralizing ROS but also by activating endogenous antioxidant defense systems through the Nrf2/ARE pathway and modulating epigenetic regulators such as HDAC1 [31,32,33,34]. However, C3G suffers from limited chemical stability under physiological pH, is highly sensitive to enzymatic degradation, and exhibits poor intestinal absorption and rapid metabolism, all of which constrain its therapeutic potential when used alone [35].

Conversely, α-tocopherol is a lipid-soluble antioxidant that preferentially accumulates in cell membranes, where it plays a critical role in quenching lipid peroxyl radicals and preventing lipid peroxidation chain reactions, which are common pathways of neuronal oxidative damage [9,36]. Importantly, α-tocopherol is more stable under physiological conditions and can potentially protect C3G from oxidative degradation, thereby enhancing its persistence and activity within biological systems.

When administered in combination, α-tocopherol may stabilize the membrane environment and preserve the functional integrity of C3G, thus prolonging its half-life and enhancing its intracellular availability. Simultaneously, C3G may augment the transcriptional activation of antioxidant genes, including HO-1 and SOD1, through epigenetic and redox-sensitive signaling mechanisms, while α-tocopherol attenuates lipid-derived oxidative stress that could otherwise dysregulate or overwhelm Nrf2 activity. This dual localization, with C3G predominantly acting in the cytosol and nucleus and α-tocopherol in the membranous compartments, creates a spatially coordinated antioxidant defense system that more effectively neutralizes ROS and prevents oxidative stress-induced damage across cellular compartments.

Furthermore, the distinct pharmacokinetics and physicochemical properties of C3G and α-tocopherol offer mutual compensation. The low bioavailability of C3G may be counterbalanced by the higher tissue retention of α-tocopherol, while the limited membrane permeability of C3G is overcome by α-tocopherol’s lipophilicity, ensuring comprehensive coverage of intracellular and membrane-associated ROS sources.

Given these findings, the combined use of C3G and α-tocopherol holds therapeutic potential for oxidative stress-related neuronal damage. Nonetheless, further studies are warranted to assess the long-term effects of HDAC1 inhibition, as well as the in vivo bioavailability and pharmacokinetics of C3G and α-tocopherol. Additionally, exploring whether this epigenetic modulation extends to other stress-responsive genes may expand the scope of its neuroprotective actions. It should also be noted that the effect of C3GE alone on HDAC1 expression under non-stressed conditions was not assessed in this study. Since the experimental design focused on the protective mechanisms during oxidative insult, the potential of C3GE to modulate HDAC1 in the absence of H_2_O_2_ remains to be elucidated. Future studies should examine whether C3GE exerts baseline epigenetic effects independent of oxidative stress.

A limitation of this study is the reliance on a relatively limited set of in vitro methodologies, including the MTT assay, ELISA, and western blotting. While these approaches provide important mechanistic insight, additional assays, such as flow cytometry, fluorescence-based ROS imaging, or quantitative PCR, would further strengthen the conclusions. Furthermore, although Nrf2 activation was assessed through western blot analysis, future studies should consider using immunocytochemistry to directly visualize Nrf2 nuclear translocation in SH-SY5Y cells. Incorporating these complementary techniques will help to validate and extend the current findings by offering spatial and quantitative resolution of key molecular events. It should also be noted that the antioxidant effects observed with 20 µg/mL and 40 µg/mL doses of C3GE were comparable, suggesting a potential saturation effect at the lower dose. This plateau may reflect a maximal activation of antioxidant pathways, beyond which higher concentrations do not yield additional benefit. Further pharmacodynamic investigations are warranted to define the optimal dose range and explore whether feedback regulation or bioavailability limitations contribute to this effect.

In conclusion, this study reveals a synergistic mechanism by which C3G, in combination with α-tocopherol, protects neuronal cells from oxidative damage. By inhibiting HDAC1, facilitating Nrf2 activation, and enhancing HO-1 and SOD1 expression, the combination improves antioxidant enzyme activities, reduces ROS and MDA levels, and restores redox balance. These findings underscore the potential of combining epigenetically active polyphenols with lipid-soluble antioxidants as a multifaceted approach to prevent or alleviate oxidative stress-induced neuronal injury. 

## 4. Materials and Methods

### 4.1. Test Substances

The non-glutinous rice variety *Oryza sativa* L. cv. RD83, officially named *Mali Dam Nong Khai 62* in Thai, was obtained from the Division of Rice Research and Development, Rice Department, Ministry of Agriculture and Cooperatives, Thailand.

The grains were thoroughly cleaned, oven-dried at 60 °C for 72 h (Memmert GmbH, Eagle, WI, USA), and ground into a fine powder. The powdered rice was then extracted by maceration using a 50% hydroalcoholic solution (ethanol: water, *v*/*v*) to enhance the yield of polar phytochemicals, particularly anthocyanins such as C3G. The extraction protocol was adapted from established methods for anthocyanin isolation from pigmented rice, which commonly employ hydroalcoholic solvents under mild acidic conditions [37,38]. The extract was centrifuged at 3000 rpm for 10 min, filtered through Whatman No. 1 filter paper, and the supernatant was concentrated using a rotary evaporator, followed by lyophilization.

α-Tocopherol (C_29_H_50_O_2_, MW 430.71), with a purity of approximately 95.5% as verified by HPLC, was purchased from Merck KGaA (Darmstadt, Germany; CAS No. 10191-41-0).

To investigate potential synergistic effects, the C3G-enriched rice extract and α-tocopherol were combined in a 1:1 ratio. This balanced proportion was selected based on preliminary dose-finding experiments, which indicated that it provided optimal antioxidant and cytoprotective effects without inducing cytotoxicity. The 1:1 ratio was also intended to ensure effective modulation of oxidative stress and epigenetic pathways by maintaining adequate concentrations of both compounds. Moreover, this approach aimed to minimize the potential risk of toxicity that could arise from higher doses of either compound when used alone. This selection was consistent with previous reports demonstrating that equimolar combinations of phenolic compounds and α-tocopherol can enhance antioxidant efficacy through complementary mechanisms [11,39].

### 4.2. HPLC Analysis of C3G

The C3G content in the *Oryza sativa* L. cv. RD83 extract was determined using HPLC equipped with a photodiode array detector (Waters^®^ 2998, Waters Corporation, Milford, MA, USA). Chromatographic separation was carried out on a Purospher^®^ STAR C18 encapped column (5 μm, 250 × 4.6 mm) with a guard cartridge (Merck KGaA). A binary solvent system consisting of 5% formic acid in water (solvent A) and methanol (solvent B) was used under gradient conditions: 15% B held for 5 min, linearly increased to 50% B over 15 min, held at 50% B for 5 min, and then returned to 15% B within 5 min for re-equilibration. The mobile phase flowed at 1.0 mL/min with a 20 μL injection volume. Absorbance was monitored at 520 nm to detect anthocyanins. The peak corresponding to C3G was identified by matching retention time and UV spectra with an authentic standard and quantified using an external calibration curve. C3G was eluted at approximately 17.8 min and the results are reported as mg/100 g of dry extract [40].

### 4.3. Determination of Total Anthocyanin Content

The total anthocyanin content of the *Oryza sativa* L. cv. RD83 extract was quantified using the pH-differential spectrophotometric method. Briefly, 1 mL of the extract was mixed separately with 2 mL of 0.025 M potassium chloride buffer (pH 1.0) and 2 mL of 0.4 M sodium acetate buffer (pH 4.5). Each mixture was incubated at 25 °C for 10 min. The absorbance of both solutions was measured at 520 and 720 nm using a UV–Visible spectrophotometer (Pharmacia LKB-Biochrom 4060, Cambridge, UK).

The anthocyanin concentration was calculated based on the difference in absorbance at the two pH values and is expressed as mg of C3G equivalents per 100 g of dry extract. The calculation was performed using a molar extinction coefficient (ε) of 26,900 L·mol^−1^·cm^−1^ and a molecular weight (MW) of 449.2 g·mol^−1^ for cyanidin-3-O-glucoside [41].

### 4.4. Antioxidant Activity Assessment

Three complementary assays, DPPH, FRAP, and ABTS, were employed to evaluate the antioxidant potential of the test samples.

#### 4.4.1. DPPH Radical Scavenging Assay

To determine the free radical scavenging capacity, samples (0–1000 µg/mL) were reacted with 0.1 mM DPPH in methanol. Following a 30 min incubation at 25 °C in the dark, the absorbance was recorded at 517 nm using a Synergy H1 Multimode Microplate Reader (BioTek Instruments, Winooski, VT, USA). Methanol was used as a blank. The results are expressed as EC_50_, the concentration needed to neutralize 50% of DPPH radicals [42,43,44].

#### 4.4.2. FRAP Assay

Reducing power was determined via the FRAP method. The working solution, freshly prepared from acetate buffer (300 mM, pH 3.6), TPTZ (10 mM), and FeCl_3_ (20 mM) in a 10:1:1 ratio, was added (190 µL) to 10 µL of the sample. After 10 min at 37 °C, the absorbance was measured at 593 nm. The EC_50_ value represented the sample’s ferric ion reducing capacity [42,43,44].

#### 4.4.3. ABTS Radical Cation Assay

ABTS•^+^ was generated by incubating 7 mM ABTS with 2.45 mM potassium persulfate for 12–16 h in the dark. A reaction mixture consisting of 30 µL of sample, 120 µL of distilled water, 30 µL of ethanol, and 3 mL of ABTS•^+^ solution was prepared. Absorbance at 734 nm was recorded using a spectrophotometer (Pharmacia LKB-Biochrom 4060). EC_50_ values were calculated to reflect radical scavenging efficiency [42,43,44].

### 4.5. Determination of Synergistic Effects

To assess the potential synergistic interaction between the C3G-enriched extract from *Oryza sativa* L. cv. RD83 and α-tocopherol, two key metrics were employed: the combination index (CI) and the dose reduction index (DRI).

#### 4.5.1. CI Value

The CI was calculated using the following equation:CI = (D_1_/EC_501_) + (D_2_/EC_502_)

In this formula, EC_501_ and EC_502_ represent the half-maximal effective concentrations (EC_50_) of each individual compound when tested alone, while D_1_ and D_2_ correspond to the concentrations of each compound when used in combination to achieve the same effect [14,15].

The interpretation of CI values was as follows:CI > 1.45 indicates antagonismCI ≈ 1 indicates an additive effectCI < 1 indicates a synergistic interaction

#### 4.5.2. DRI Value

The DRI quantifies the extent to which the effective dose of a compound can be reduced when used in combination, compared to its use alone, while still achieving a comparable biological response. The DRI was determined using the equation:DRI = EC_50_ (alone)/D (in combination)

Here, EC_50_ (alone) is the concentration required to produce 50% of the maximum effect when the compound is administered individually, and D (in combination) is the dose required to produce the same effect when used in conjunction with another compound [14,15].

Together, the CI and DRI provided a quantitative evaluation of the interactive effects between the C3G-enriched extract from *Oryza sativa* L. cv. RD83 and α-tocopherol, offering insight into their potential synergistic antioxidant efficacy.

### 4.6. Cell Culture

The SH-SY5Y cell line, originally derived from human neuroblastoma and widely recognized for its dopaminergic neuronal properties, was sourced from the American Type Culture Collection (ATCC, CRL-2266; Manassas, VA, USA). Cells were cultured in Dulbecco’s modified Eagle’s medium (DMEM; Thermo Fisher Scientific, Waltham, MA, USA) fortified with 10% fetal bovine serum (FBS), along with 1% penicillin-streptomycin and 1% non-essential amino acids to support optimal growth. Cultivation was carried out in a controlled environment maintained at 37 °C with 5% CO_2_ and high humidity. Before initiating experimental treatments, the existing culture medium was withdrawn and substituted with fresh medium supplemented with H_2_O_2_, either alone or in combination with test substances, depending on the experimental condition [14,15,40,41].

### 4.7. Cell Viability Assay

To evaluate cytotoxicity and assess neuroprotective efficacy, the MTT assay was utilized on SH-SY5Y neuroblastoma cells. Cells were seeded into 96-well plates at a density of 1 × 10^4^ cells per well and incubated under standard culture conditions until confluence. Initially, cells were exposed to a range of H_2_O_2_ concentrations (0–800 µM) in serum-free DMEM for 24 h to establish the optimal dose required for inducing oxidative stress. The effective H_2_O_2_ concentration for subsequent experiments was selected based on previous data [14,15,40,41].

To investigate the protective potential of test substances, SH-SY5Y cells were pre-incubated with varying concentrations of the combination (0–1280 µg/mL) for 24 h, a duration selected based on previous studies demonstrating that a 24 h exposure is optimal for cellular uptake, antioxidant response activation, and minimizing delayed cytotoxicity under oxidative stress conditions [14,15]. Following this pretreatment, the medium was replaced with fresh serum-free DMEM containing H_2_O_2_, with or without the test compounds, and cells were incubated for an additional hour at 37 °C under 5% CO_2_ [14,15].

Subsequently, MTT reagent (0.5 mg/mL; MilliporeSigma, St. Louis, MO, USA) was added to each well, and the plates were incubated for 4 h to allow for formazan formation. After incubation, the reagent was removed, and the resulting crystals were solubilized using 100 µL of dimethyl sulfoxide (DMSO). Absorbance was measured at 570 nm using a Synergy H1 Multimode Microplate Reader (BioTek Instruments). Cell viability was expressed as a percentage relative to the untreated control [14,15].

### 4.8. Evaluation of Oxidative Stress Biomarkers

#### 4.8.1. Lipid Peroxidation

Lipid peroxidation levels were evaluated by estimating MDA, a final product formed during the oxidative degradation of lipids. A modified TBARS assay protocol was applied, in which 50 µL of the cellular lysate was mixed with 8.1% sodium dodecyl sulfate (SDS), 0.8% thiobarbituric acid (TBA), 20% acetic acid, and deionized water to achieve a total reaction volume of 1 mL. All chemicals used were of analytical grade and purchased from MilliporeSigma. The mixtures were incubated at 95 °C for 1 h to promote MDA–TBA complex formation. After cooling under running tap water, the chromogenic product was extracted with 1.25 mL of a butanol-pyridine solution (15:1, *v*/*v*; Merck KGaA). Following centrifugation at 4000 rpm for 10 min, the supernatant was carefully removed, and the absorbance was recorded at 532 nm using a spectrophotometer (Pharmacia LKB-Biochrom 4060). A calibration curve was constructed using 1,1,3,3-tetramethoxypropane (TMP), and MDA levels are reported as nanograms per milligram of protein [45].

#### 4.8.2. Intracellular Reactive Oxygen Species

To monitor ROS accumulation within cells, a fluorescent probe, CM-H_2_DCFDA (5-(and-6)-carboxy-2′,7′-dichlorofluorescein diacetate), was utilized. This non-fluorescent compound permeates cell membranes and is hydrolyzed by intracellular esterases. Upon oxidation by ROS, it yields a green, fluorescent product, dichlorofluorescein (DCF). SH-SY5Y cells were seeded in black-walled 96-well plates and exposed to the test substances for 24 h. After treatment, cells were incubated with 10 µM CM-H_2_DCFDA for 30 min at 37 °C in a dark, humidified CO_2_ incubator. Post-incubation, cells were washed with phosphate-buffered saline (PBS) and challenged with H_2_O_2_ under serum-free conditions for an additional 24 h. Fluorescence intensity, indicative of ROS levels, was measured using a Synergy H1 Multimode Reader (BioTek Instruments) at 488 nm excitation and 520 nm emission wavelengths.

Together, MDA quantification and ROS detection provided a comprehensive analysis of cellular oxidative burden and facilitated the evaluation of the protective properties of the treatment under investigation [46].

### 4.9. Assessment of Antioxidant Enzyme Activities

To evaluate the enzymatic defense mechanisms against oxidative stress, the activities of CAT, SOD, and GSH-Px were quantified in SH-SY5Y cell lysates following treatment.

#### 4.9.1. CAT Activity

CAT activity was analyzed by its capacity to decompose H_2_O_2_, with the residual H_2_O_2_ reacting with potassium permanganate (KMnO_4_). The assay involved combining 10 µL of the cell extract with 50 µL of 30 mM H_2_O_2_ (prepared in 50 mM phosphate buffer, pH 7.0), followed by 150 µL of 5 mM KMnO_4_ and 25 µL of 4 M H_2_SO_4_. After vortexing, the absorbance was measured at 490 nm using a Synergy H1 Microplate Reader (BioTek Instruments). A standard curve was prepared using purified CAT enzyme (MilliporeSigma) at known concentrations (0–100 U/mL). The results were normalized to protein content and are expressed as units per milligram of protein [47].

#### 4.9.2. SOD Activity

SOD activity was determined based on its ability to inhibit the reduction of cytochrome C by superoxide radicals generated in a xanthine–xanthine oxidase system. The reaction solution included 0.5 mM xanthine, 0.01 M EDTA, 15 mM cytochrome C, and 0.2 M phosphate buffer (pH 7.8) mixed in a ratio of 50:1:1:25 (*v*/*v*). A 200 µL aliquot of this mixture was incubated with 20 µL of cell lysate and 20 µL of xanthine oxidase (0.90 mU/mL). Absorbance changes were monitored at 415 nm using the same reader. Calibration was performed with standard SOD (MilliporeSigma) ranging from 0 to 25 U/mL. Data are expressed as units per milligram of protein [48].

#### 4.9.3. GSH-Px Activity

GSH-Px activity was evaluated by monitoring the oxidation of glutathione (GSH) in the presence of H_2_O_2_. The assay mixture included 100 µL of 40 mM KH_2_PO_4_ buffer (pH 7.0), 10 µL of 1 mM dithiothreitol (DTT), 10 µL of 50 mM GSH, 10 µL of 10 mM monosodium phosphate, and 100 µL of 30% H_2_O_2_. After adding 20 µL of the cell extract, the mixture was incubated at 25 °C for 10 min. Then, 10 µL of 10 mM DTNB was added, and the absorbance at 412 nm was recorded. GSH-Px standards form MilliporeSigma (0–5 U/mL) were used for calibration. Enzyme activity is reported as units per milligram of protein [49].

### 4.10. Western Blotting Analysis

To evaluate protein expression levels of HDAC1, Nrf2, HO-1, and SOD1, western blotting was performed on SH-SY5Y cell lysates. Cells were lysed in ice-cold N-PER™ neuronal protein extraction buffer (Thermo Fisher Scientific), which is optimized for neural tissue protein recovery. Lysates were centrifuged at 10,000× *g* for 10 min at 4 °C, and the resulting supernatant was collected for protein quantification using a NanoDrop 2000c spectrophotometer (Thermo Fisher Scientific).

Equal amounts of total protein (60 µg) were mixed with Tris-Glycine SDS sample buffer and denatured by heating at 95 °C for 5 min. Samples were separated on SDS-polyacrylamide gels and transferred onto nitrocellulose membranes (Bio-Rad, Hercules, CA, USA). Membranes were blocked with 5% non-fat dry milk in Tris-buffered saline containing 0.1% Tween-20 (TBS-T) for 1 h at room temperature.

Following blocking, membranes were incubated overnight at 4 °C with the following primary antibodies diluted 1:1000: anti-HDAC1 (Cat. No. 34589, Cell Signaling Technology, Danvers, MA, USA), anti-Nrf2 (Cat. No. 12721, Cell Signaling Technology), anti-HO-1 (Cat. No. 43966, Cell Signaling Technology), and anti-SOD1 (Cat. No. 2770, Cell Signaling Technology). Anti-β-actin (Cat. No. 4967S, Cell Signaling Technology; 1:2000) was used as a loading control.

After washing with 0.05% TBS-T, membranes were incubated with HRP-conjugated anti-rabbit IgG secondary antibody (Cell Signaling Technology; 1:2000) for 1 h at room temperature. Signal detection was carried out using Immobilon Forte Western HRP substrate (Merck KGaA), and chemiluminescent bands were visualized using the ChemiDoc™ MP Imaging System (Bio-Rad). Band intensities were quantified using Image Lab software (version 6.0.0, Bio-Rad), normalized to β-actin expression, and expressed relative to the untreated control group. Full-length blots are provided in the Appendix A (Appendix A: protein loading; Appendix A: HDAC1 expression; Appendix A: Nrf2 expression; Appendix A: HO-1 expression; Appendix A: SOD1 expression; Appendix A: β-actin expression), and representative cropped images are included in the main figures [50,51].

### 4.11. Statistical Analysis

Quantitative results are expressed as the mean ± SEM. Group differences were evaluated using one-way ANOVA, followed by Tukey’s multiple comparison test when more than two groups were involved. Significance was defined as a *p*-value less than 0.05. Data processing and statistical calculations were performed using SPSS Statistics (v21.0; IBM Corp., Armonk, NY, USA). 

## 5. Conclusions

This study demonstrates that the combined treatment of C3GE exerts a synergistic antioxidant effect against H_2_O_2_-induced oxidative stress in SH-SY5Y cells. The co-treatment not only restored cell viability but also significantly enhanced antioxidant enzyme activities, reduced intracellular ROS, and suppressed lipid peroxidation. Mechanistically, C3GE promoted the downregulation of HDAC1 and the activation of the Nrf2/HO-1 and SOD1 signaling axis, highlighting an epigenetically mediated antioxidant response. These findings suggest that the synergistic interplay between hydrophilic and lipophilic antioxidants confers robust neuroprotection by integrating direct radical scavenging with transcriptional and epigenetic regulation. This combinatorial approach offers promising potential for the development of functional neuroprotective agents targeting oxidative stress-related neurodegenerative disorders. Further in vivo investigations and clinical studies are warranted to validate its therapeutic efficacy and mechanistic relevance in more complex biological systems.

## Figures and Tables

**Figure 1 ijms-26-06490-f001:**
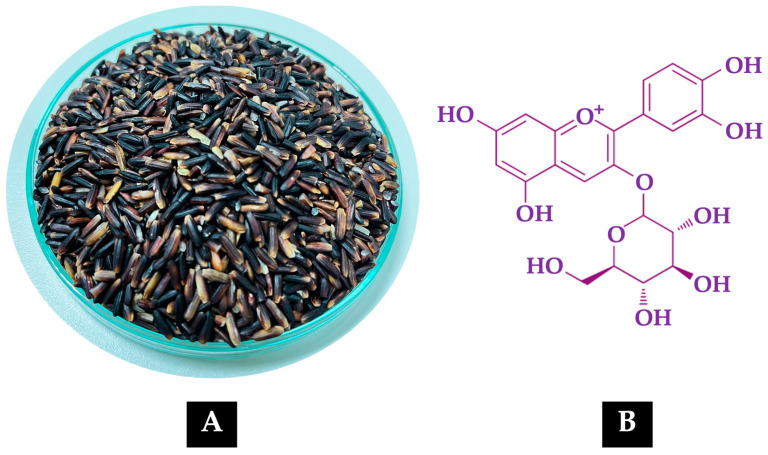
(**A**) Grains of the non-glutinous purple rice variety *Oryza sativa* L. cv. RD83 (*Mali Dam Nong Khai 62*). (**B**) Molecular structure of C3G, the predominant anthocyanin found in its extract.

**Figure 2 ijms-26-06490-f002:**
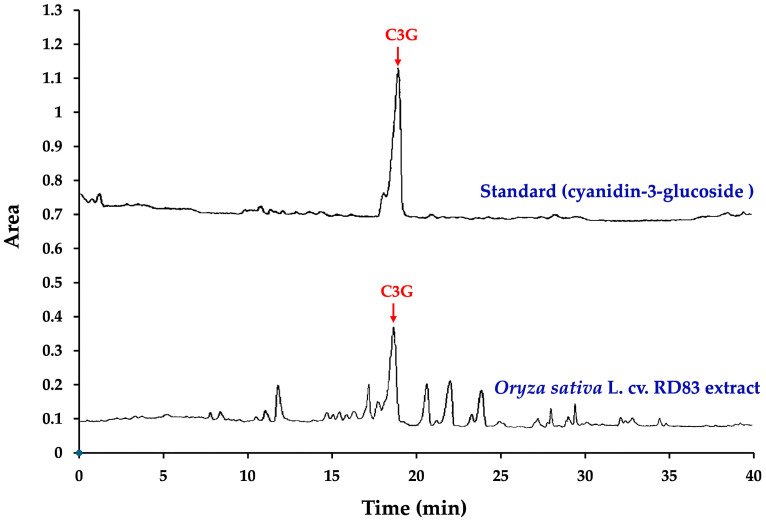
Chromatographic fingerprint of C3G in the extract of *Oryza sativa* L. cv. RD83, obtained by HPLC.

**Figure 3 ijms-26-06490-f003:**
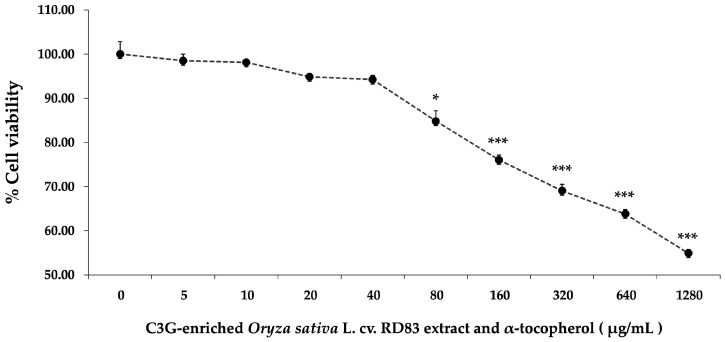
Cytotoxicity of C3GE on SH-SY5Y cell viability. Data are expressed as mean ± SEM (*n* = 8). * *p* < 0.05, *** *p* < 0.001 vs. naïve control (0 µg/mL). C3GE refers to the combination of C3G-enriched extract from *Oryza sativa* L. cv. RD83 and α-tocopherol.

**Figure 4 ijms-26-06490-f004:**
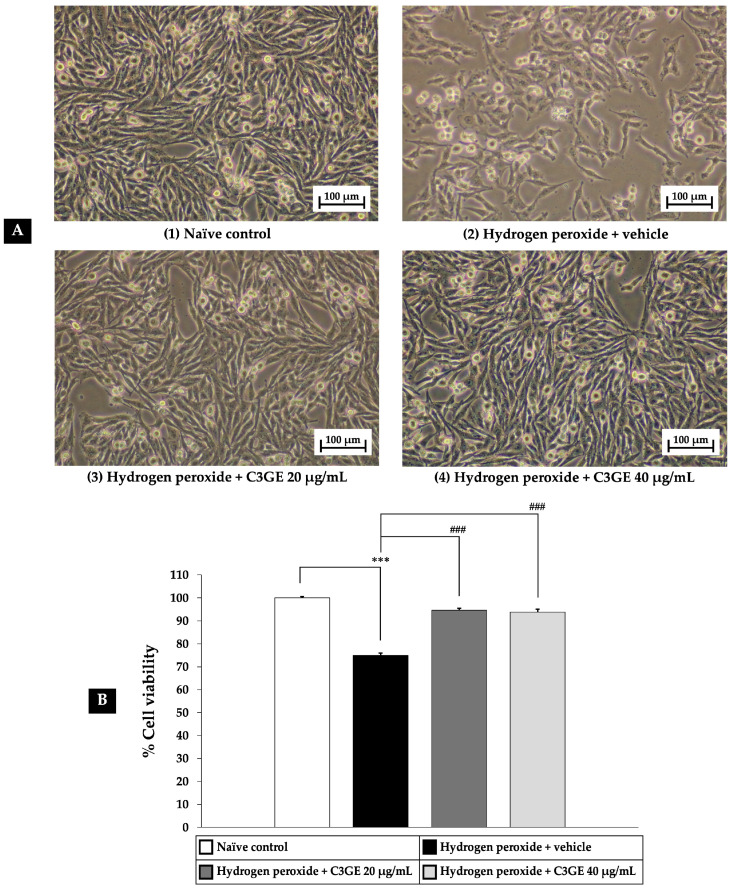
Cytoprotective effects of C3GE against H_2_O_2_-induced oxidative stress in SH-SY5Y cells. (**A**) Representative images showing cell density under 10× magnification. (**B**) Quantitative analysis of cell viability. Data are expressed as mean ± SEM (*n* = 8). *** *p* < 0.001 vs. naïve control; ^###^ *p* < 0.001 vs. H_2_O_2_ plus vehicle-treated group. H_2_O_2_: 200 µM hydrogen peroxide; C3GE refers to the combination of C3G-enriched extract from *Oryza sativa* L. cv. RD83 and α-tocopherol.

**Figure 5 ijms-26-06490-f005:**
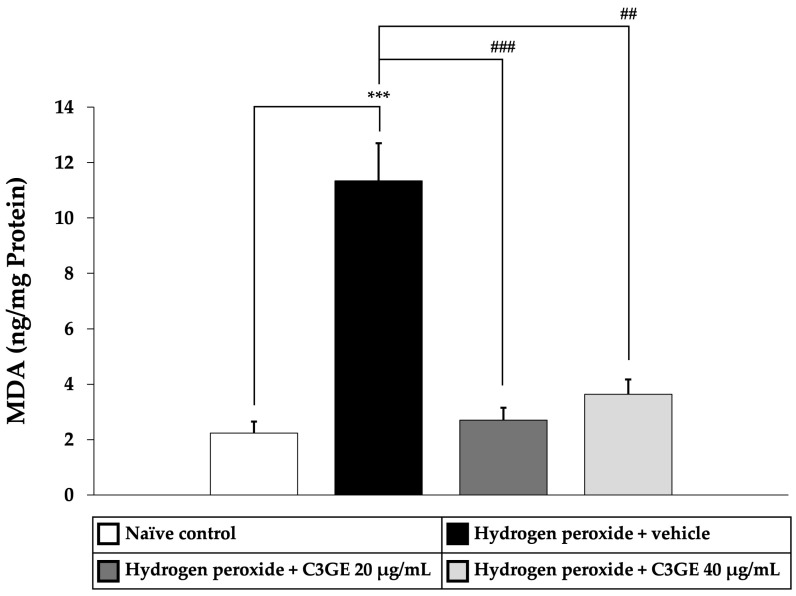
Effect of C3GE on MDA levels in SH-SY5Y cells exposed to H_2_O_2_-induced oxidative stress. Data are expressed as mean ± SEM (*n* = 4). *** *p* < 0.001 vs. naïve control; ^##^ *p* < 0.01, ^###^ *p* < 0.001 vs. H_2_O_2_ plus vehicle-treated group. H_2_O_2_: 200 µM hydrogen peroxide. C3GE refers to the combination of C3G-enriched extract from *Oryza sativa* L. cv. RD83 and α-tocopherol.

**Figure 6 ijms-26-06490-f006:**
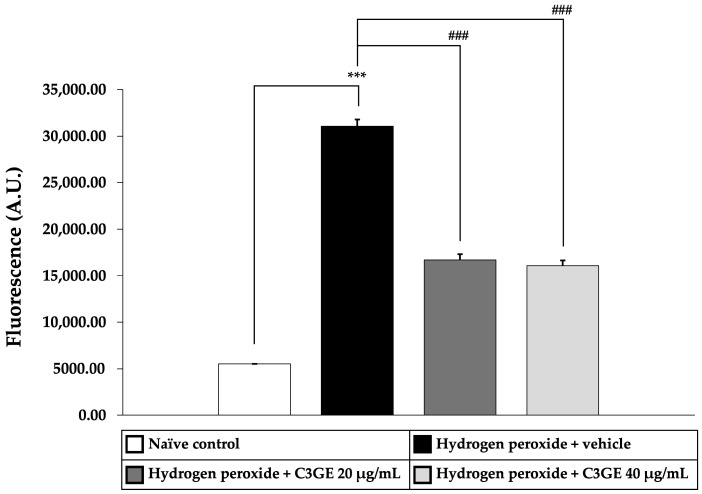
Effect of C3GE on ROS production in SH-SY5Y cells exposed to H_2_O_2_-induced oxidative stress. Data are expressed as mean ± SEM (*n* = 4). *** *p* < 0.001 vs. naïve control; ^###^ *p* < 0.001 vs. H_2_O_2_ plus vehicle-treated group. H_2_O_2_: 200 µM hydrogen peroxide. C3GE refers to the combination of C3G-enriched extract from *Oryza sativa* L. cv. RD83 and α-tocopherol.

**Figure 7 ijms-26-06490-f007:**
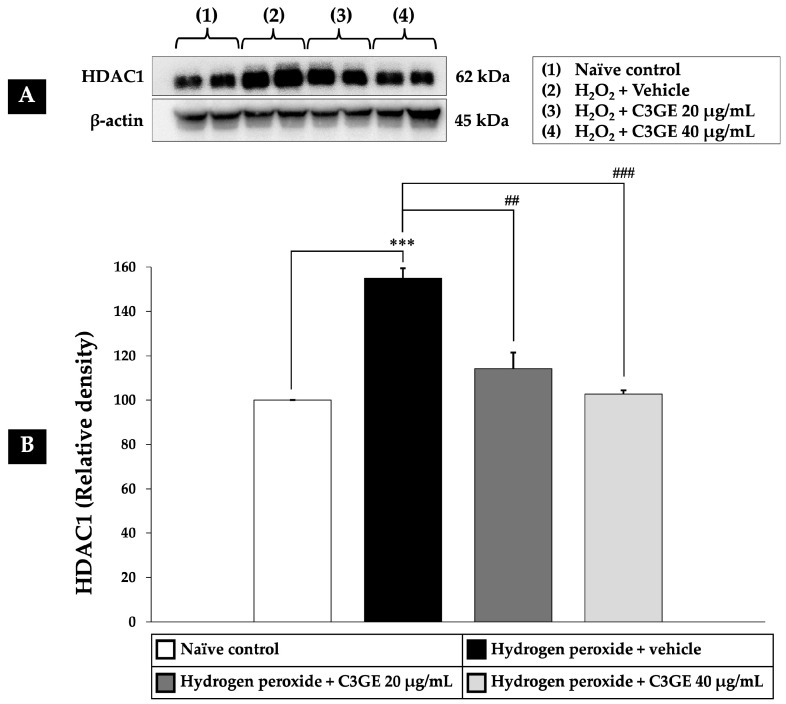
Effect of C3GE on HDAC1-mediated epigenetic modulation in in SH-SY5Y cells exposed to H_2_O_2_-induced oxidative stress. (**A**) Representative immunoblot showing HDAC1 protein expression. (**B**) Quantification of HDAC1 relative to β-actin. Data are presented as mean ± SEM (*n* = 4). *** *p* < 0.001 vs. naïve control; ^##^ *p* < 0.01, ^###^ *p* < 0.001 vs. H_2_O_2_ plus vehicle-treated group. H_2_O_2_: 200 µM hydrogen peroxide. C3GE refers to the combination of C3G-enriched extract from *Oryza sativa* L. cv. RD83 and α-tocopherol.

**Figure 8 ijms-26-06490-f008:**
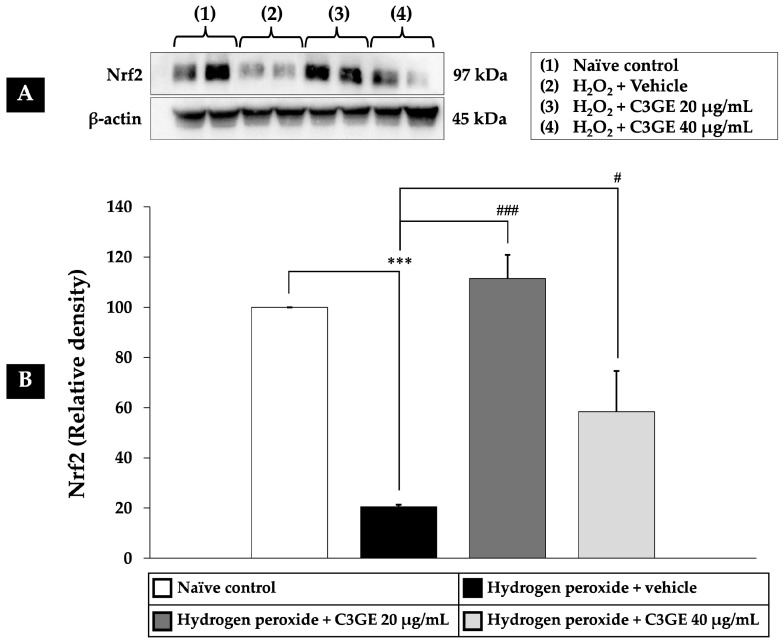
Effect of C3GE on Nrf2 expression in SH-SY5Y cells exposed to H_2_O_2_-induced oxidative stress. (**A**) Representative immunoblot showing Nrf2 protein expression. (**B**) Quantification of Nrf2 relative to β-actin. Data are presented as mean ± SEM (*n* = 4). *** *p* < 0.001 vs. naïve control; ^#^ *p* < 0.05, ^###^ *p* < 0.001 vs. H_2_O_2_ plus vehicle-treated group. H_2_O_2_: 200 µM hydrogen peroxide. C3GE refers to the combination of C3G-enriched extract from *Oryza sativa* L. cv. RD83 and α-tocopherol.

**Figure 9 ijms-26-06490-f009:**
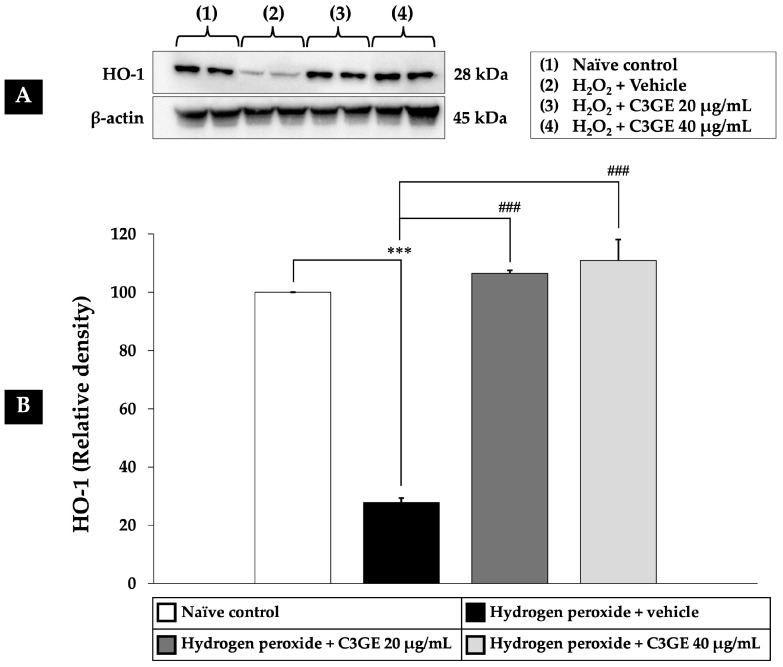
Effect of C3GE on HO-1 expression in SH-SY5Y cells exposed to H_2_O_2_-induced oxidative stress. (**A**) Representative immunoblot showing HO-1 protein expression. (**B**) Quantification of HO-1 relative to β-actin. Data are presented as mean ± SEM (*n* = 4). *** *p* < 0.001 vs. naïve control; ^###^ *p* < 0.001 vs. H_2_O_2_ plus vehicle-treated group. H_2_O_2_: 200 µM hydrogen peroxide. C3GE refers to the combination of C3G-enriched extract from *Oryza sativa* L. cv. RD83 and α-tocopherol.

**Figure 10 ijms-26-06490-f010:**
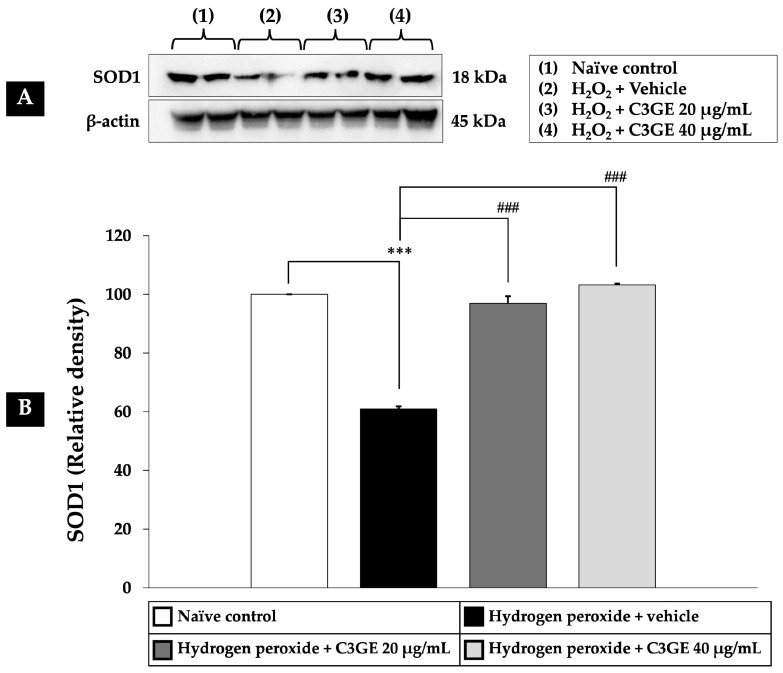
Effect of C3GE on SOD1 expression in SH-SY5Y cells exposed to H_2_O_2_-induced oxidative stress. (**A**) Representative immunoblot showing SOD1 protein expression. (**B**) Quantification of SOD1 relative to β-actin. Data are presented as mean ± SEM (*n* = 4). *** *p* < 0.001 vs. naïve control; ^###^ *p* < 0.001 vs. H_2_O_2_ plus vehicle-treated group. H_2_O_2_: 200 µM hydrogen peroxide. C3GE refers to the combination of C3G-enriched extract from *Oryza sativa* L. cv. RD83 and α-tocopherol.

**Figure 11 ijms-26-06490-f011:**
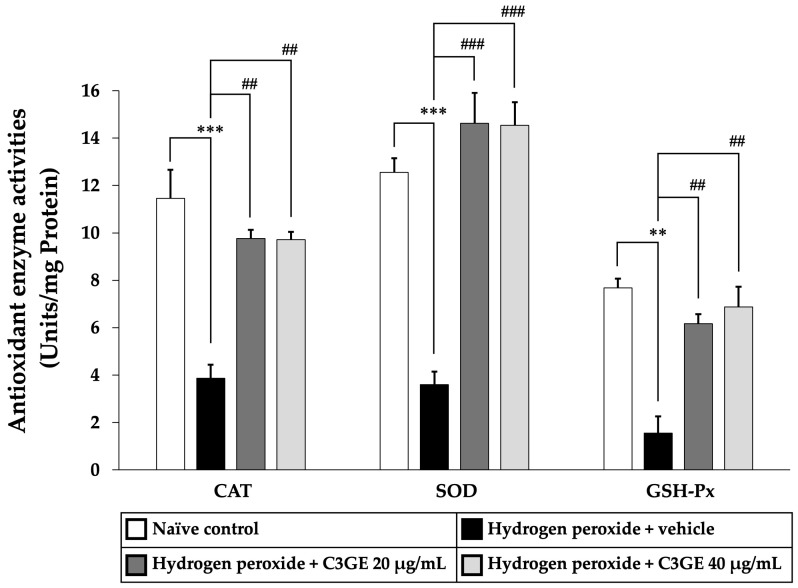
Effect of C3GE on antioxidant enzyme activities in SH-SY5Y cells exposed to H_2_O_2_-induced oxidative stress. Data are expressed as mean ± SEM (*n* = 4). ** *p* < 0.01, *** *p* < 0.001 vs. naïve control; ^##^ *p* < 0.01, ^###^ *p* < 0.001 vs. H_2_O_2_ plus vehicle-treated group. H_2_O_2_: 200 µM hydrogen peroxide. C3GE refers to the combination of C3G-enriched extract from *Oryza sativa* L. cv. RD83 and α-tocopherol.

**Table 1 ijms-26-06490-t001:** Anthocyanin content and antioxidant activities of *Oryza sativa* L. cv. RD83 extract, α-tocopherol, and C3GE.

Parameters	Units	*Oryza sativa* L. cv. RD83 Extract	α-Tocopherol	C3GE
Total anthocyanin content	mg C3G/100 g dry extract	527.00 ± 8.82	-	-
C3G	mg/100 g dry extract	364.44 ± 0.01	-	-
Antioxidant activities				
DPPH	EC_50_ (μg/mL)	43.31 ± 1.72	17.16 ± 1.13	6.29 ± 0.51 ***^,##^
FRAP	EC_50_ (μg/mL)	45.09 ± 1.66	26.77 ± 1.16	11.58 ± 1.64 ***^,##^
ABTS	EC_50_ (μg/mL)	83.14 ± 1.49	18.35 ± 0.74	9.85 ± 0.19 ***^,##^

Data are presented as mean ± SEM (*n* = 3). *** *p* < 0.001, comparison between *Oryza sativa* L. cv. RD83 extract and C3GE. ^##^ *p* < 0.01, comparison between α-tocopherol and C3GE. C3GE refers to the combination of C3G-enriched extract from *Oryza sativa* L. cv. RD83 and α-tocopherol.

**Table 2 ijms-26-06490-t002:** Combination index (CI) and dose reduction index (DRI) values of C3GE.

Parameters	Combination Index (Type of Interaction)	Dose Reduction Index	
		*Oryza sativa* L. cv. RD83 Extract	α-Tocopherol
Antioxidant activities			
DPPH	0.52 ± 0.04 (synergism)	7.04 ± 0.53	2.79 ± 0.22
FRAP	0.69 ± 0.06 (synergism)	4.11 ± 0.45	2.42 ± 0.23
ABTS	0.66 ± 0.02 (synergism)	8.44 ± 0.02	1.87 ± 0.06

Data are presented as mean ± SEM (*n* = 3).

## Data Availability

The original contributions presented in this study are included in the article and Appendix A. Further inquiries can be directed to the corresponding authors.

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
