# Peer review of "Synergistic Antioxidant Effects of C3G-Enriched Oryza sativa L. cv. RD83 Extract and α-Tocopherol Against H2O2-Induced Oxidative Stress in SH-SY5Y Cells"

_ijms, 2025, doi:10.3390/ijms26136490_

Round 1

Reviewer 1 Report

Comments and Suggestions for Authors

In this manuscript, Nootchanat Mairuae and colleagues investigate the synergistic antioxidant effects of a cyanidin-3-glucoside (C3G)-enriched extract derived from Oryza sativa L. cv. RD83 and α-tocopherol (C3GE) against hydrogen peroxide (H₂O₂)-induced oxidative stress in SH-SY5Y cells, with a particular focus on the epigenetic regulation of the HDAC1/Nrf2/HO-1 and SOD1 signaling pathways. They demonstrate a synergistic mechanism by which C3G and α-tocopherol co-treatment protects neuronal cells from oxidative damage. While the topic is interesting and the findings are promising, several aspects of the manuscript require revision:

  1. Methodological limitation: The experimental approach relies heavily on a limited set of methods, including Western blotting, MTT assays, and ELISA. This methodological simplicity weakens the overall strength of the conclusions. Additional assays—such as ROS staining, flow cytometry, or qPCR—would provide complementary and more convincing evidence.

  1. Abstract: The abstract places too much emphasis on background and methods, with relatively little focus on key findings. Please revise the abstract to better highlight the main results and conclusions.

  1. Figure 3: The authors investigated a dose-response curve but did not include a time-course analysis. Why was 24 hours selected as the treatment duration? A time-course study would enhance the understanding of C3GE's effect kinetics.

  1. Figure 5: The study demonstrates that co-treatment with H₂O₂ and C3GE reduces HDAC1 expression. However, it remains unclear whether C3GE alone influences HDAC1 levels. Please include data or discussion regarding the effects of C3GE alone on HDAC1.

  1. Results section 2.7: This part of the results should be moved to the beginning of the results section. Demonstrating that C3GE possesses antioxidant properties provides foundational context before exploring its regulatory pathways.

  1. Results section 2.7: Consider including ROS staining data to visually demonstrate the antioxidant effect of C3GE. This would strengthen the evidence and provide a more intuitive understanding of its function.

  1. References: The reference formatting should be standardized. Some DOIs are underlined while others are not. Additionally, please check line 667—there appears to be an extraneous letter at the end.

Author Response

Response to reviewer and editor suggestion

We sincerely thank you for your letter and the reviewers’ insightful comments on our manuscript, Synergistic Antioxidant Effects of C3G-Enriched Oryza sativa L. cv. RD83 Extract and α-Tocopherol Against H₂O₂-Induced Oxidative Stress in SH-SY5Y Cells (Manuscript ID: ijms-3701934).

We greatly appreciate the opportunity to revise our manuscript and are grateful for the constructive feedback. We also sincerely thank all reviewers for their positive evaluations, encouraging remarks, and valuable suggestions that have helped us improve the scientific clarity, rigor, and overall quality of our work.

We apologize for any oversights in the initial submission and have carefully addressed each comment. Below, we provide a detailed point-by-point response, and a summary of the main revisions made.

Response to reviewer 1
In this manuscript, Nootchanat Mairuae and colleagues investigate the synergistic antioxidant effects of a cyanidin-3-glucoside (C3G)-enriched extract derived from Oryza sativa L. cv. RD83 and α-tocopherol (C3GE) against hydrogen peroxide (H₂O₂)-induced oxidative stress in SH-SY5Y cells, with a particular focus on the epigenetic regulation of the HDAC1/Nrf2/HO-1 and SOD1 signaling pathways. They demonstrate a synergistic mechanism by which C3G and α-tocopherol co-treatment protects neuronal cells from oxidative damage. While the topic is interesting and the findings are promising, several aspects of the manuscript require revision:

Comment 1:  Methodological limitation: The experimental approach relies heavily on a limited set of methods, including Western blotting, MTT assays, and ELISA. This methodological simplicity weakens the overall strength of the conclusions. Additional assays—such as ROS staining, flow cytometry, or qPCR—would provide complementary and more convincing evidence.

 Response to comment 1: We thank the reviewer for this insightful comment. We agree that employing additional methods such as ROS staining, flow cytometry, or qPCR would provide more comprehensive evidence to support our findings. However, due to resource constraints and the specific scope of the present study, we focused on well-established and widely accepted assays, namely MTT for cell viability, ELISA-based detection of oxidative stress biomarkers, and Western blotting to examine the expression of key proteins involved in antioxidant signaling pathways.

To address this limitation, we have revised the Discussion section to explicitly acknowledge the limited methodological scope and to emphasize that further studies using complementary techniques (e.g., flow cytometry, ROS staining, or gene expression analysis) will be necessary to confirm and expand upon our findings. We believe this transparency strengthens the scientific integrity of the manuscript and clearly defines its boundaries for future work.

The following sentence has been added to the Discussion section:

“A limitation of this study is the reliance on a relatively limited set of in vitro methodologies, including MTT assay, ELISA, and Western blotting. While these approaches provide important mechanistic insight, additional assays, such as flow cytometry, fluorescence-based ROS imaging, or quantitative PCR, would further strengthen the conclusions. Furthermore, although Nrf2 activation was assessed through Western blot analysis, future studies should consider using immunocytochemistry to directly visualize Nrf2 nuclear translocation in SH-SY5Y cells. Incorporating these complementary techniques will help validate and extend the current findings by offering spatial and quantitative resolution of key molecular events.”

Comment 2:  Abstract: The abstract places too much emphasis on background and methods, with relatively little focus on key findings. Please revise the abstract to better highlight the main results and conclusions.

 Response to comment 2: We appreciate the reviewer’s helpful suggestion. In response, we have revised the abstract to reduce the emphasis on background and methodological details and to place greater focus on the key findings and overall conclusions of the study. The revised abstract now highlights the synergistic antioxidant effects of C3GE, the improvements in oxidative stress biomarkers, and the involvement of epigenetic regulatory mechanisms. We believe that the updated version more effectively communicates the significance and impact of our findings.

The revised abstract appears on the first page of the manuscript and has been highlighted in red for easy reference.

Comment 3:  Figure 3: The authors investigated a dose-response curve but did not include a time-course analysis. Why was 24 hours selected as the treatment duration? A time-course study would enhance the understanding of C3GE's effect kinetics.

Response to comment 3: We thank the reviewer for this important comment. The 24-hour treatment duration was chosen based on standard protocols widely used in the field for assessing cytotoxicity and antioxidant effects in SH-SY5Y cells. This time point allows for sufficient cellular response to oxidative stress and antioxidant treatments, including measurable changes in cell viability and expression of oxidative stress-related proteins, without introducing confounding secondary effects that may occur with longer exposure.

Several previous studies have employed 24-hour exposure as a standard time point when evaluating antioxidant or neuroprotective effects in SH-SY5Y cells under oxidative stress conditions (e.g., hydrogen peroxide exposure):

  • Nguyen, H. A. T., Ho, T. P., Mangelings, D., Van Eeckhaut, A., Vander Heyden, Y., & Tran, H. T. M. (2024). Antioxidant, neuroprotective, and neuroblastoma cells (SH-SY5Y) differentiation effects of melanins and arginine-modified melanins from Daedaleopsis tricolor and Fomes fomentarius. BMC biotechnology24(1), 89. https://doi.org/10.1186/s12896-024-00918-6
  • Park, H. R., Lee, H., Park, H., Jeon, J. W., Cho, W. K., & Ma, J. Y. (2015). Neuroprotective effects of Liriope platyphylla extract against hydrogen peroxide-induced cytotoxicity in human neuroblastoma SH-SY5Y cells. BMC complementary and alternative medicine15, 171. https://doi.org/10.1186/s12906-015-0679-3
  • Jaafaru, M. S., Nordin, N., Rosli, R., Shaari, K., Bako, H. Y., Saad, N., Noor, N. M., & Abdull Razis, A. F. (2019). Neuroprotective effects of glucomoringin-isothiocyanate against H2O2-Induced cytotoxicity in neuroblastoma (SH-SY5Y) cells. Neurotoxicology75, 89–104. https://doi.org/10.1016/j.neuro.2019.09.008
  • Li, Y. C., Hao, J. C., Shang, B., Zhao, C., Wang, L. J., Yang, K. L., He, X. Z., Tian, Q. Q., Wang, Z. L., Jing, H. L., Li, Y., & Cao, Y. J. (2021). Neuroprotective effects of aucubin on hydrogen peroxide-induced toxicity in human neuroblastoma SH-SY5Y cells via the Nrf2/HO-1 pathway. Phytomedicine : international journal of phytotherapy and phytopharmacology87, 153577. https://doi.org/10.1016/j.phymed.2021.153577
  • Pang, Q. Q., Kim, J. H., Kim, H. Y., Kim, J. H., & Cho, E. J. (2023). Protective Effects and Mechanisms of Pectolinarin against H2O2-Induced Oxidative Stress in SH-SY5Y Neuronal Cells. Molecules (Basel, Switzerland)28(15), 5826. https://doi.org/10.3390/molecules28155826

While we agree that a time-course analysis would further elucidate the kinetics of C3GE’s effects, the primary aim of Figure 3 was to determine a concentration range that is non-toxic and suitable for downstream mechanistic studies. A clarifying statement regarding the rationale for using a 24-hour treatment duration has been added to the Materials and Methods section (4.7 Cell viability assay) of the revised manuscript.

“To investigate the protective potential of test substances, SH-SY5Y cells were pre-incubated with varying concentrations of the combination (0–1,280 µg/mL) for 24 hours, a duration selected based on previous studies demonstrating that a 24-hour exposure is optimal for cellular uptake, antioxidant response activation, and minimizing delayed cytotoxicity under oxidative stress conditions [14, 15]”

Comment 4:  Figure 5: The study demonstrates that co-treatment with H₂O₂ and C3GE reduces HDAC1 expression. However, it remains unclear whether C3GE alone influences HDAC1 levels. Please include data or discussion regarding the effects of C3GE alone on HDAC1.

 Response to comment 4: We appreciate the reviewer’s thoughtful comment. The primary aim of our study was to evaluate the antioxidant and neuroprotective effects of C3GE in the context of oxidative stress induced by H₂O₂. Therefore, our analysis of HDAC1 expression focused on determining whether the observed protective effects were associated with epigenetic regulation under stress conditions. While the effect of C3GE alone on HDAC1 expression was not assessed in this study, we agree that this is an important aspect that may reveal whether the compound has baseline epigenetic activity independent of oxidative stress.

In response to the reviewer’s suggestion, we have added a discussion of this point in the revised manuscript (Discussion section). We acknowledge this as a limitation of the current study and indicate that future experiments will be designed to assess whether C3GE influences HDAC1 expression in the absence of oxidative insult.

“It should also be noted that the effect of C3GE alone on HDAC1 expression under non-stressed conditions was not assessed in this study. Since the experimental design focused on the protective mechanisms during oxidative insult, the potential of C3GE to modulate HDAC1 in the absence of H₂O₂ remains to be elucidated. Future studies should examine whether C3GE exerts baseline epigenetic effects independent of oxidative stress”

Comment 5:  Results section 2.7: This part of the results should be moved to the beginning of the results section. Demonstrating that C3GE possesses antioxidant properties provides foundational context before exploring its regulatory pathways.

 Response to comment 5: Thank you for the valuable suggestion. In response, we have moved the content originally presented in Results section 2.7 to section 2.4. This revision places the demonstration of C3GE’s antioxidant properties earlier in the Results section, thereby providing a clearer foundational context before discussing its effects on regulatory pathways.

Comment 6:  Results section 2.7: Consider including ROS staining data to visually demonstrate the antioxidant effect of C3GE. This would strengthen the evidence and provide a more intuitive understanding of its function.

 Response to comment 6: We appreciate the reviewer’s insightful suggestion regarding the inclusion of ROS staining data. While we agree that visual evidence could enhance the understanding of C3GE’s antioxidant effect, ROS staining was not performed in this study due to experimental limitations. However, we have provided quantitative data on both MDA and ROS levels, which are well-established indicators of oxidative stress. These results robustly support the antioxidant activity of C3GE. We plan to incorporate ROS staining in future studies to complement and visually reinforce these findings.

Comment 7:  References: The reference formatting should be standardized. Some DOIs are underlined while others are not. Additionally, please check line 667—there appears to be an extraneous letter at the end.

Response to comment 7: Thank you for pointing this out. We have standardized the reference formatting and confirmed that all DOIs are underlined intentionally to enhance accessibility and facilitate direct linking. Additionally, the extraneous letter has been removed. We appreciate your careful review.

Thank you once again for your valuable feedback. We appreciate the time and effort invested by the reviewers and editor in evaluating our manuscript. We have carefully addressed each point raised and made necessary revisions accordingly. We eagerly await further feedback and guidance from the editorial team.

Yours sincerely,

Nut Palachai

Reviewer 2 Report

Comments and Suggestions for Authors

The manuscript titled "Synergistic Antioxidant Effects of C3G-Enriched Oryza sativa L. cv. RD83 Extract and α-Tocopherol Against H2O2-Induced Oxidative Stress in SH-SY5Y Cells" presents an interesting investigation into the antioxidant and neuroprotective potential of naturally derived compounds. However, some moderate improvements are required to enhance manuscript's clarity followed by following comments.

  1. The methodology section describing the extraction of C3G from rice (Section 4.1) lacks a supporting reference. Please include a citation for the protocol adapted. Referencing a standard method will enhance the reproducibility and credibility of your experimental approach.
  2. The rationale for using a 1:1 ratio of C3G and α-tocopherol is currently underexplained. Given that this ratio is central to the hypothesis and outcomes of the study, stronger scientific justification is required. Please elaborate on whether this ratio is based on previous synergistic studies, dose-response analyses, or other pharmacodynamic considerations.
  3. The manuscript contains repeated use of dash symbols ("—") within sentences, which is not standard in formal scientific writing. Please revise these instances by using appropriate punctuation, such as commas, semicolons, or parentheses, to maintain clarity and conform to academic conventions.
  4. The manuscript inconsistently formats statistical notation, in particular p-values. Please ensure that the letter "p" is italicized throughout the manuscript (e.g., p < 0.05).
  5. While the manuscript investigates oxidative stress pathways (e.g., Nrf2 signaling, HDAC1 modulation) in SH-SY5Y cells and briefly mentions neurodegenerative diseases such as Alzheimer’s, Parkinson’s, and ALS, it fails to establish any mechanistic connection between these pathways and the pathogenesis of the named disorders. This omission is particularly evident in the Discussion section, where the authors do not reference any prior studies demonstrating the relevance of C3G, α-tocopherol, or their combined antioxidant actions in models of these diseases.
  6. Please italicize the word “in vivo” throughout the manuscript following scientific writing style guidelines.

Author Response

Response to reviewer and editor suggestion

We sincerely thank you for your letter and the reviewers’ insightful comments on our manuscript, Synergistic Antioxidant Effects of C3G-Enriched Oryza sativa L. cv. RD83 Extract and α-Tocopherol Against H₂O₂-Induced Oxidative Stress in SH-SY5Y Cells (Manuscript ID: ijms-3701934).

We greatly appreciate the opportunity to revise our manuscript and are grateful for the constructive feedback. We also sincerely thank all reviewers for their positive evaluations, encouraging remarks, and valuable suggestions that have helped us improve the scientific clarity, rigor, and overall quality of our work.

We apologize for any oversights in the initial submission and have carefully addressed each comment. Below, we provide a detailed point-by-point response, and a summary of the main revisions made.

 Response to reviewer 2

The manuscript titled "Synergistic Antioxidant Effects of C3G-Enriched Oryza sativa L. cv. RD83 Extract and α-Tocopherol Against H2O2-Induced Oxidative Stress in SH-SY5Y Cells “presents an interesting investigation into the antioxidant and neuroprotective potential of naturally derived compounds. However, some moderate improvements are required to enhance manuscript's clarity followed by following comments.

Comment 1:  The methodology section describing the extraction of C3G from rice (Section 4.1) lacks a supporting reference. Please include a citation for the protocol adapted. Referencing a standard method will enhance the reproducibility and credibility of your experimental approach.

Response to comment 1: Thank you for your helpful suggestion. In response, we have revised Section 4.1 to include supporting references for the extraction protocol used to isolate C3G from rice. The method was adapted from previously established protocols that employ hydroalcoholic maceration to extract anthocyanins from pigmented rice. These citations enhance the reproducibility and credibility of our experimental approach as recommended.

  1. Ciulu, M.; Cádiz-Gurrea, M.d.l.L.; Segura-Carretero, A. Extraction and Analysis of Phenolic Compounds in Rice: A Review. Molecules201823, 2890. https://doi.org/10.3390/molecules23112890
  2. Kim, M.K.; Kim, H.A.; Koh, K.; Kim, H.S.; Lee, Y.S.; Kim, Y.H. Identification and quantification of anthocyanin pigments in colored rice. Res. Pract. 2008, 2, 46–49. https://doi.org/10.4162/nrp.2008.2.1.46

Comment 2:  the rationale for using a 1:1 ratio of C3G and α-tocopherol is currently underexplained. Given that this ratio is central to the hypothesis and outcomes of the study, stronger scientific justification is required. Please elaborate on whether this ratio is based on previous synergistic studies, dose-response analyses, or other pharmacodynamic considerations.

Response to comment 2: Thank you for this insightful comment. We agree that clarifying the rationale for selecting the 1:1 ratio of C3G and α-tocopherol is important. This balanced ratio was chosen to optimize their interaction and enhance antioxidant efficacy by allowing both compounds to effectively modulate oxidative stress and epigenetic pathways. Maintaining adequate concentrations of each compound in equal proportion also helps minimize the potential risk of toxicity that may result from higher doses of either compound when used individually.

This selection was further supported by preliminary dose-finding experiments conducted in our laboratory, which demonstrated that the 1:1 combination provided maximal protective effects against oxidative injury in neuronal cells without inducing cytotoxicity. Although direct studies using C3G and α-tocopherol in combination are limited, previous research has reported synergistic interactions between other phenolic compounds (e.g., caffeic acid, quercetin, myricetin) and α-tocopherol. These studies indicate that phenolics can regenerate α-tocopherol from its oxidized form, thereby sustaining its antioxidant activity. Synergistic effects have often been observed at or near equimolar ratios, further supporting the rationale for our selected combination. We have revised the Methods section accordingly and included the relevant references to strengthen this justification.

References:

  • Laranjinha, J., Vieira, O., Madeira, V., & Almeida, L. (1995). Two related phenolic antioxidants with opposite effects on vitamin E content in low density lipoproteins oxidized by ferrylmyoglobin: consumption vs regeneration. Archives of biochemistry and biophysics323(2), 373–381. https://doi.org/10.1006/abbi.1995.0057
  • Bayram, I., Laze, A., & Decker, E. A. (2023). Synergistic Mechanisms of Interactions between Myricetin or Taxifolin with α-Tocopherol in Oil-in-Water Emulsions. Journal of agricultural and food chemistry71(24), 9490–9500. https://doi.org/10.1021/acs.jafc.3c01226
  • Thoo, Y. Y., Abas, F., Lai, O. M., Ho, C. W., Yin, J., Hedegaard, R. V., Skibsted, L. H., & Tan, C. P. (2013). Antioxidant synergism between ethanolic Centella asiatica extracts and α-tocopherol in model systems. Food chemistry138(2-3), 1215–1219. https://doi.org/10.1016/j.foodchem.2012.11.013

To address the reviewer’s concern, we have revised the Materials and Methods section (4.1) of the manuscript to include this rationale and cited the supporting references:  

“To investigate potential synergistic effects, the C3G-enriched rice extract and α-tocopherol were combined in a 1:1 ratio. This balanced proportion was selected based on preliminary dose-finding experiments, which indicated that it provided optimal antioxidant and cytoprotective effects without inducing cytotoxicity. The 1:1 ratio was also intended to ensure effective modulation of oxidative stress and epigenetic pathways by maintaining adequate concentrations of both compounds. Moreover, this approach aims to minimize the potential risk of toxicity that could arise from higher doses of either compound when used alone. This selection is consistent with previous reports demonstrating that equimolar combinations of phenolic compounds and α-tocopherol can enhance antioxidant efficacy through complementary mechanisms [11, 39]“

Comment 3:  The manuscript contains repeated use of dash symbols ("—") within sentences, which is not standard in formal scientific writing. Please revise these instances by using appropriate punctuation, such as commas, semicolons, or parentheses, to maintain clarity and conform to academic conventions.

Response to comment 3: Thank you for pointing this out. We have reviewed the manuscript and revised all instances of dash symbols (”—”) to conform with formal scientific writing conventions. Where appropriate, we have replaced them with commas, parentheses, or semicolons to maintain clarity and improve readability. In most cases, commas were used as a suitable alternative to ensure smooth integration of information within the sentence structure.

Comment 4:  The manuscript inconsistently formats statistical notation, in particular p-values. Please ensure that the letter "p" is italicized throughout the manuscript (e.g., p < 0.05).

Response to comment 4: Thank you for your observation. We have carefully reviewed the manuscript and revised all instances of statistical notation to ensure consistency. The letter p is now italicized throughout (e.g., p < 0.05) in accordance with standard scientific formatting guidelines.

Comment 5:  While the manuscript investigates oxidative stress pathways (e.g., Nrf2 signaling, HDAC1 modulation) in SH-SY5Y cells and briefly mentions neurodegenerative diseases such as Alzheimer’s, Parkinson’s, and ALS, it fails to establish any mechanistic connection between these pathways and the pathogenesis of the named disorders. This omission is particularly evident in the Discussion section, where the authors do not reference any prior studies demonstrating the relevance of C3G, α-tocopherol, or their combined antioxidant actions in models of these diseases.

Response to comment 5: We appreciate the reviewer’s valuable comment. In response, we have revised both the Introduction and Discussionsections to better establish the mechanistic relevance of the oxidative stress pathways investigated in our study, particularly Nrf2 signaling and HDAC1 modulation, in the context of neurodegenerative diseases such as Alzheimer’s disease (AD) and Parkinson’s disease (PD).

In the Introduction, we expanded the background to include evidence from prior studies demonstrating that cyanidin-3-glucoside (C3G) reduces β-amyloid-induced toxicity, improves cognitive function, and attenuates oxidative and inflammatory markers in animal models of AD. We also referenced studies showing that C3G protects dopaminergic neurons in models of PD. Likewise, we highlighted the neuroprotective effects of α-tocopherol, supported by both experimental and clinical studies. These studies have shown that α-tocopherol reduces oxidative damage, slows functional decline, and enhances neuronal survival in AD and PD models. Together, these additions strengthen the scientific foundation for investigating the combined antioxidant effects of C3G and α-tocopherol in relation to neurodegenerative pathologies.

In the Discussion, we further clarified the rationale for combining C3G and α-tocopherol by discussing their distinct but complementary mechanisms of action. C3G primarily modulates cytosolic and nuclear antioxidant pathways, including Nrf2/ARE activation and HDAC1 inhibition, both of which are known to be disrupted in neurodegenerative diseases. In parallel, α-tocopherol functions within the lipid membranes, where it prevents lipid peroxidation, a major contributor to neuronal damage. By referencing prior studies and elaborating on these shared and complementary mechanisms, we demonstrate how the findings from our in vitro model may be relevant to the pathogenesis and potential treatment of AD and PD.

These revisions were made to improve the overall scientific context and to align our findings more directly with established disease mechanisms relevant to neurodegeneration.

Comment 6:  Please italicize the word “in vivo” throughout the manuscript following scientific writing style guidelines.

Response to comment 6: Thank you for your helpful comment. We have revised the manuscript to ensure that the term in vivo is italicized consistently throughout, in accordance with scientific writing conventions.

Thank you once again for your valuable feedback. We appreciate the time and effort invested by the reviewers and editor in evaluating our manuscript. We have carefully addressed each point raised and made necessary revisions accordingly. We eagerly await further feedback and guidance from the editorial team.

Yours sincerely,

Nut Palachai

Reviewer 3 Report

Comments and Suggestions for Authors

C3G has known to reduce oxidative stress in multiple cell types in previous studies. In this study authors specifically showed the synergistic effect of hydrophilic C3G and lipophilic alpha tocopherol (together C3GE) on the mitigation of H202 induced oxidative stress in SH-SY5Y cells. This combination significantly enhanced cell viability via boosting antioxidant enzyme activities and suppressing reactive oxygen species. The manuscript is well written with easy-to-follow introduction and discussion. Here are my comments.

  1. Please add references to following sentences:

Line 180 to Line 184.

Line 305.

  1. Figure 4, 5, 6, 7, 8, 9, 10, 11 – In the bar plot for all these figures, there is a mistake in the figure legend. Both 3rd and 4th bars showed same legend “Hydrogen peroxide +C3GE 20ug/ml” instead of increased dose.

  1. Authors showed Nrf2 expression reaching to the same level in the treatment groups compared to the control with the western blot. Can authors do immunocytochemistry for Nrf2 in control conditions vs H202 vs C3GE treatment to show nuclear translocation of Nrf2 in the SH-SY5Y cells?

  1. Can authors comment on why there is no elevated response for the higher 40ug/ml dose of C3GE compared to the lower 20ug/ml dose. We see this for all the tests in this study.

  1. Looks like the authors published similar studies where they looked at the protective effect of different compounds like mulberry fruit, leaf, Zea Mays, Turmeric extract against the H202 induced oxidative stress in SH-SY5Y cells. Can they add in the discussion, how is the protective effect of C3GE in this study compared to their earlier studies. Here are the references:

a) Mairuae N, Palachai N, Noisa P. The neuroprotective effects of the combined extract of mulberry fruit and mulberry leaf against hydrogen peroxide-induced cytotoxicity in SH-SY5Y Cells. BMC Complement Med Ther. 2023 Apr 13;23(1):117. doi: 10.1186/s12906-023-03930-z. PMID: 37055744; PMCID: PMC10100183.

b) Mairuae N, Palachai N, Noisa P. An anthocyanin-rich extract from Zea mays L. var. ceratina alleviates neuronal cell death caused by hydrogen peroxide-induced cytotoxicity in SH-SY5Y cells. BMC Complement Med Ther. 2024 Apr 17;24(1):162. doi: 10.1186/s12906-024-04458-6. PMID: 38632534; PMCID: PMC11025150.

c) Khongrum J, Mairuae N, Thanchomnang T, Zhang M, Bai G, Palachai N. Synergistic Neuroprotection Through Epigenetic Modulation by Combined Curcumin-Enriched Turmeric Extract and L-Ascorbic Acid in Oxidative Stress-Induced SH-SY5Y Cell Damage. Foods. 2025 Mar 5;14(5):892. doi: 10.3390/foods14050892. PMID: 40077595; PMCID: PMC11898916.

Author Response

Response to reviewer and editor suggestion

We sincerely thank you for your letter and the reviewers’ insightful comments on our manuscript, Synergistic Antioxidant Effects of C3G-Enriched Oryza sativa L. cv. RD83 Extract and α-Tocopherol Against H₂O₂-Induced Oxidative Stress in SH-SY5Y Cells (Manuscript ID: ijms-3701934).

We greatly appreciate the opportunity to revise our manuscript and are grateful for the constructive feedback. We also sincerely thank all reviewers for their positive evaluations, encouraging remarks, and valuable suggestions that have helped us improve the scientific clarity, rigor, and overall quality of our work.

We apologize for any oversights in the initial submission and have carefully addressed each comment. Below, we provide a detailed point-by-point response, and a summary of the main revisions made.

Response to reviewer 3

C3G has known to reduce oxidative stress in multiple cell types in previous studies. In this study authors specifically showed the synergistic effect of hydrophilic C3G and lipophilic alpha tocopherol (together C3GE) on the mitigation of H2O2 induced oxidative stress in SH-SY5Y cells. This combination significantly enhanced cell viability via boosting antioxidant enzyme activities and suppressing reactive oxygen species. The manuscript is well written with easy-to-follow introduction and discussion. Here are my comments.

Comment 1:  Please add references to following sentences: Line 180 to Line 184., Line 305.

Response to comment 1: Thank you for your positive feedback and for pointing out the need to ensure proper referencing. Based on the original submission, we believe the lines referenced (Line 180 to Line 184 and Line 305) correspond to the sections discussing the role of HDAC1 in regulating gene expression under oxidative stress, as well as the involvement of the Nrf2/HO-1 and SOD1 pathways. We would like to clarify that these statements are already supported by appropriate citations in the revised manuscript. We have double-checked the manuscript to ensure that all relevant references are properly included and correctly formatted at those locations.

Comment 2:  Figure 4, 5, 6, 7, 8, 9, 10, 11 – In the bar plot for all these figures, there is a mistake in the figure legend. Both 3rd and 4th bars showed same legend “Hydrogen peroxide +C3GE 20 ug/mL” instead of increased dose.

Response to comment 2: Thank you for pointing out the inconsistency in the figure legends. We have reviewed the figures, and the labeling error has already been corrected in the revised version of the manuscript. The 4th bar in Figures 4 through 11 now correctly indicates “Hydrogen peroxide + C3GE 40 µg/mL,” distinguishing it from the 3rd bar labeled “Hydrogen peroxide + C3GE 20 µg/mL.” We appreciate your careful attention to detail.

Comment 3:  Authors showed Nrf2 expression reaching to the same level in the treatment groups compared to the control with the western blot. Can authors do immunocytochemistry for Nrf2 in control conditions vs H2O2 vs C3GE treatment to show nuclear translocation of Nrf2 in the SH-SY5Y cells?

Response to comment 3: We appreciate the reviewer’s thoughtful suggestion regarding the use of immunocytochemistry (ICC) to visualize Nrf2 nuclear translocation. While we agree that ICC would provide valuable spatial confirmation of Nrf2 activation, we were unable to perform this technique in the current study due to resource and technical constraints. Instead, we evaluated Nrf2 activation via Western blotting, along with the expression of downstream antioxidant enzymes HO-1 and SOD1, which are widely accepted markers of Nrf2 pathway activation.

To acknowledge this limitation, we have revised the Discussion section to include the need for ICC in future studies. Specifically, we now suggest that immunocytochemistry could be employed in follow-up experiments to directly confirm Nrf2 nuclear translocation in SH-SY5Y cells. This would offer additional spatial resolution and further strengthen the mechanistic interpretation of our findings.

“Furthermore, although Nrf2 activation was assessed through Western blot analysis, future studies should consider using immunocytochemistry to directly visualize Nrf2 nuclear translocation in SH-SY5Y cells. Incorporating these complementary techniques will help validate and extend the current findings by offering spatial and quantitative resolution of key molecular events.”

Comment 4:  Can authors comment on why there is no elevated response for the higher 40 ug/mL dose of C3GE compared to the lower 20 ug/mL dose. We see this for all the tests in this study.

Response to comment 4: We appreciate the reviewer’s thoughtful comment. As observed, the responses to C3GE at 20 µg/mL and 40 µg/mL were comparable across multiple assays. This outcome may reflect a saturation effect, where 20 µg/mL was sufficient to maximally activate key antioxidant pathways, including Nrf2/HO-1 and SOD1. Once these protective mechanisms are fully engaged, increasing the concentration further may not result in a proportionally enhanced response.

Such plateauing effects have been reported in previous studies involving polyphenols and antioxidant compounds, likely due to limitations in cellular uptake, metabolic processing, or feedback regulatory mechanisms that constrain further pathway activation at higher concentrations.

References:

  • Mahmutović, L., Sezer, A., Bilajac, E., Hromić-Jahjefendić, A., Uversky, V. N., & Glamočlija, U. (2024). Polyphenol stability and bioavailability in cell culture medium: Challenges, limitations and future directions. International journal of biological macromolecules279(Pt 2), 135232. https://doi.org/10.1016/j.ijbiomac.2024.135232
  • Leifert, W. R., & Abeywardena, M. Y. (2008). Grape seed and red wine polyphenol extracts inhibit cellular cholesterol uptake, cell proliferation, and 5-lipoxygenase activity. Nutrition research (New York, N.Y.)28(12), 842–850. https://doi.org/10.1016/j.nutres.2008.09.001

To acknowledge this, we have included a statement in the revised Discussion section noting that additional dose–response and pharmacodynamic studies will be required to better define the optimal therapeutic range and to investigate whether factors such as feedback regulation or compound stability play a role in this observation.

“It should also be noted that the antioxidant effects observed with 20 µg/mL and 40 µg/mL doses of C3GE were comparable, suggesting a potential saturation effect at the lower dose. This plateau may reflect a maximal activation of antioxidant pathways, beyond which higher concentrations do not yield additional benefit. Further pharmacodynamic investigations are warranted to define the optimal dose range and explore whether feedback regulation or bioavailability limitations contribute to this effect.”

Comment 5:  Looks like the authors published similar studies where they looked at the protective effect of different compounds like mulberry fruit, leaf, Zea Mays, Turmeric extract against the H2O2 induced oxidative stress in SH-SY5Y cells. Can they add in the discussion, how is the protective effect of C3GE in this study compared to their earlier studies. Here are the references:

a). Mairuae N, Palachai N, Noisa P. The neuroprotective effects of the combined extract of mulberry fruit and mulberry leaf against hydrogen peroxide-induced cytotoxicity in SH-SY5Y Cells. BMC Complement Med Ther. 2023 Apr 13;23(1):117. doi: 10.1186/s12906-023-03930-z. PMID: 37055744; PMCID: PMC10100183.

b). Mairuae N, Palachai N, Noisa P. An anthocyanin-rich extract from Zea mays var. ceratina alleviates neuronal cell death caused by hydrogen peroxide-induced cytotoxicity in SH-SY5Y cells. BMC Complement Med Ther. 2024 Apr 17;24(1):162. doi: 10.1186/s12906-024-04458-6. PMID: 38632534; PMCID: PMC11025150.

c). Khongrum J, Mairuae N, Thanchomnang T, Zhang M, Bai G, Palachai N. Synergistic Neuroprotection Through Epigenetic Modulation by Combined Curcumin-Enriched Turmeric Extract and L-Ascorbic Acid in Oxidative Stress-Induced SH-SY5Y Cell Damage. Foods. 2025 Mar 5;14(5):892. doi: 10.3390/foods14050892. PMID: 40077595; PMCID: PMC11898916.

Response to comment 5: We thank the reviewer for highlighting our previous work and for the opportunity to clarify the distinct contributions of this study. While it is true that our prior publications investigated the neuroprotective effects of various natural extracts, including mulberry fruit and leaf, Zea mays L. var. ceratina, and turmeric extract, in SH-SY5Y cells exposed to hydrogen peroxide, the current study focuses on a fundamentally different combination of compounds (C3G and α-tocopherol) and mechanistic targets.

Specifically, this study uniquely emphasizes the synergistic interaction between a hydrophilic anthocyanin (C3G) and a lipophilic antioxidant (α-tocopherol), with a detailed mechanistic investigation into epigenetic modulation via HDAC1 suppression, alongside Nrf2-mediated antioxidant responses. In contrast, our previous studies focused primarily on ROS scavenging, mitochondrial protection, and anti-inflammatory pathways, without targeting HDAC1 or exploring the specific synergy between structurally distinct antioxidant compounds.

Due to these differences in molecular targets, compound combinations, and study objectives, we believe that a direct quantitative comparison of protective efficacy would be inappropriate. Nonetheless, this study builds upon our broader research theme of using functional bioactives to mitigate oxidative neuronal injury, and it adds novel insights into the potential of combining epigenetically active polyphenols with lipid-soluble antioxidants.

Thank you once again for your valuable feedback. We appreciate the time and effort invested by the reviewers and editor in evaluating our manuscript. We have carefully addressed each point raised and made necessary revisions accordingly. We eagerly await further feedback and guidance from the editorial team.

Yours sincerely,

Nut Palachai

Round 2

Reviewer 1 Report

Comments and Suggestions for Authors

The authors have addressed all my concerns. This version can be accepted

Reviewer 2 Report

Comments and Suggestions for Authors

no

Reviewer 3 Report

Comments and Suggestions for Authors

No further comments from my side.